# Information sharing and deferral option in cybersecurity investment

**Chuanxi Cai** [ID]**¹\***, **Liurong Zhao²**

**1** School of Economic and Management, Nanjing Forest University, Jiang Su Province, China, **2** School of Economic and Management, Nanjing Tech University, Jiang Su Province, China

\* ccx@njfu.edu.cn

**Data Availability Statement:** All relevant data are within the paper.

**Funding:** The Science and Technology Innovation Fund (163060171) and the General Program in philosophy and Social Sciences (2022SJYB0124)

## Abstract

This study investigates the effect of information sharing and deferral option on a firm's information security investment strategies by considering strategic interactions between a firm and an attacker. We find that 1) information sharing decreases a firm's security investment rate. 2) If a deferral decision is possible, the firm will decrease its immediate investment, and avoid non-investment. 3) After information sharing, the probability of a firm's deferral decision increases for low-benefit information ($S_L$) but decreases for high-benefit information ($S_H$). 4) When information sharing accuracy is low, a firm only defers decisions in a fraction of $S_L$; when information sharing accuracy is high, the firm defers its decisions in all $S_L$ and a fraction of $S_H$. 5) Information sharing can improve the effect of deferral decision when accuracy is low but weaken it when accuracy is high. These results contradict the literature, wherein information sharing reduces a firm's uncertainty on cybersecurity investment and decreases deferment options associated with investment.

## 1. Introduction

Corporations worldwide are currently making critical investments in various cybersecurity-related activities. Hence, security investments in information systems have become critical in information security economics. Hausken [1] believes substantial security investment is needed to deter most perpetrators. Gao and Zhong [2] analysed information security investment strategies under both targeted and mass attacks by considering strategic interactions between two competitive firms and a hacker. Qian et al. [3] determined a new game of information sharing and security investment between two allied firms. Considering information security insurance, Qian et al. [4] determined an information security investment game between two firms with complementary information assets. Shao et al. [5] analysed the impact of reputational concerns on information security managers' investment decisions. Li Xiaotong [6] provided a solution for the information security investment decisions of complementary enterprises under the characteristics of multi-enterprise and non-cooperative enterprises. Li Xiaotong [7] conducted an evolutionary game-theoretical analysis of enterprise information security investments based on an information-sharing platform. Li and Xue [8] conducted an economic analysis of information security investment decisions for substitutable enterprises.

are all received by Chuanxi Cai. The Science and Technology Innovation Fund (163060171) is supported by Nanjing Forestry University, and the General Program in philosophy and Social Sciences (2022SJYB0124) is supported by Jiang Su Provice. There is no additional external funding received for this study The funders had no role in study design, data collection and analysis, decision to publish, or preparation of the manuscript.

**Competing interests:** The authors have declared that no competing interests exist.

The potential benefits (payoffs) of security investments largely involve potential cost savings, which are riddled with significant uncertainty. Therefore, to obtain more information, firms tend to defer much of their cybersecurity investments unless they are reacting to a major breach [9]. Moreover, when the firms obtain information about the intrusion of an attacker, they can share the information to decrease the cost of their defence. Bian et al. [10] indicated that investors are prone to imitate their neighbours' activities through a comprehensive analysis of the risk dominance degree of certain investment behaviour. Overall, before making cybersecurity investments, firms tend obtain more accurate information through information sharing or decrease uncertainty by deferring decisions.

Deferral (postponing one decision to a later date) is a recent occurrence in cybersecurity investment; however considerable research has been made in psychology [11,12] and management [13,14] literature. White et al. [11] suggested that choice deferrals can arise from absolute evaluations or relative comparisons. Larasati and Yeh [13] demonstrated that a more attractive choice always decreases choice deferral. Berens and Funke [15] determined how situational and personal factors influence two different forms of decision avoidance: 1) deferring choice to a later point in time (decision deferral) and 2) refusing both alternatives (option refusal). Deferral can lead to better decisions by enabling a search for additional information or better alternatives. However, this can be risky because a cybersecurity breach could occur during the deferral process. In this study, the deferral option is one of the firm's strategies before making cybersecurity investments. Hence, the firm has three strategies for cybersecurity investment: immediate investment, no investment, and decision-making after deferral. If the firm cannot make decision between investment and no investment, it can select the deferral option. That is, the firm can make no decision for some time and decide whether to invest after obtaining more information by deferring. Previous research on deferral has mainly focused on factors affecting the deferral decision-making of firms but this article focus on the effect of deferral option on a firm's expected benefits.

Moreover, information sharing has a certain history in cybersecurity investment [3,16–23], which can help firms obtain more useful information about the attacker and firm. However, information sharing is also associated with the free-rider problem, risk of information leakage and information errors [24]. Therefore, incentives for information sharing are harder to furnish. Information sharing between firms is a deterrent to an attack, which could change the choice of the attacker [23]. Hausken [25] analysed a firm's proactive and retroactive defence against hackers with information sharing in four-period games. Hausken [25] only considered information sharing between hackers and not information sharing between firms. Gordon et al. [26] demonstrated how information sharing could encourage firms to proactively (and not reactively) approach cybersecurity investments. Assuming that information obtained through information sharing is accurate, Gordon et al. [26] believe that a firm's information sharing decreases the value of its deferment option associated with the investment. In Gordon et al. [26], a firm's cost saving from information security investment by hiring a cybersecurity consulting firm—according to a Chief Security Officer (CSO)—would be either $40000 (low-saving) or $200000 (high-saving) a month, and with an identical probability (50%). However, practically, the probabilities cannot be equal as they are affected by the attacker's decision. Additionally, Gordon et al. [26] assumed that the accuracy of the information-sharing signal was fixed and equal between low-saving and high-saving signals. Hence, Chief Financial Officer (CFO) has no choice but to invest in the CSO's high-saving signal or defer their decision in the CSO's low-saving signal. However, in practice, the CSO's strategy is affected by mood, hunger, stress, sleep deprivation, risk preference, etc., and accuracy is neither equal nor fixed. Additionally, the CFO can close investments without deferring.

Motivated by Gordon et al. [26] in information security investment, we construct a model between a firm and attacker, and analyse the effect of a firm's information sharing and deferral option on a firm's expected benefits. Our model reflects the probability of the firm's cost saving, accuracy of the information-sharing signal, and the CFO's different decision. For example, regardless of whether an attacker chooses to intrude, the CSO can obtain a high-saving signal or a low-saving signal. Regardless of the type of the signal the CFO receives from the CSO, the CFO has three strategies: immediate investment, no investment, and decision-making after deferral. Similar to the proactive and retroactive defences in Hausken [25], a firm's defence is proactive if it invests immediately and retroactive if it invests later after a deferral. Unlike the analysis of the interplay between the information sharing of hackers and the defence strategies of firms in Hausken [25], this study examines the effect of information sharing and deferral option on a firm's expected benefits. Moreover, for convenience, we assume that information accuracy after information sharing is only affected by the CSO's risk preference. Hence, we did not consider the CSO's mood, hunger, stress, sleep deprivation, and other elements. An interesting result in this study is that information sharing can improve the effect of deferral decision on a firm's expected benefits when accuracy of information is low, but it weakens the effect when information sharing accuracy is high. This interesting result contradicts Gordon et al. [26] wherein information sharing decreases the value of the deferment option associated with investment.

The remainder of this paper is organized as follows. Section 2 establishes a game model of the interaction between a firm and attacker. The model describes both the firm's deferral option and information sharing. Section 3 derives the equilibrium strategies and the primary results of the model. Section 4 derives the results on the value of the deferral option and information sharing under the default and optimal conditions respectively, and then presents the simulation and analysis of the results. Finally, Section 5 concludes the study.

## 2. Model

### 2.1. Model description

This study assumes that a firm has joined an industry-specific information-sharing group. No charges are incurred for joining this information-sharing group, providing that the firm is willing to share cybersecurity-related information with the group's members (i.e., free-riders are excluded from this group) [26]. Based on the agreement, all firms report detailed information to the group's members on their actual cybersecurity breaches and the steps taken to prevent and respond to cybersecurity breaches. Hence, the study constructs a game model between a firm and an attacker; however, the firm can enjoy information sharing. In this study, information sharing occurs between the members of the industry-specific information-sharing group. Additionally, the analysis in this study concerns information sharing between firms, not information sharing between attackers. The firm now has three strategies: immediate investment, no investment, and decision-making after deferral. An attacker only has two strategies: intrusion and no intrusion. In this study, the deferral option constitutes the firm's security investment strategy, not the strategy of the firm's information sharing. That is, firm's security investment can be deferred, firm's information sharing cannot be deferred. Table 1 list the notations used in our subsequent discussion.

When an attacker intrudes, the firm can learn more about the attacker if it invests in defense. If a firm invests in defense and the attacker does not intrude, the firm cannot benefit. Hence, the firm benefits if the firm invests and the attacker intrudes. When a firm invests and an attacker intrudes, we assume that the cost of a firm's investment is $C$, the benefit of the firm's investment is $H$, and the attacker will have a benefit of $\mu$ but incur penalty of $\beta$. That is,

**Table 1. List of notations (all the probability is between 0 and 1).**

| Parameters | Explanation |
|---|---|
| $H$ | Benefits to the firm when a firm invests and an attacker intrudes ($H>C$) |
| $L$ | Benefits to the firm when a firm invests and an attacker does not intrude ($L<C$) |
| $C$ | Costs of each investment for a firm |
| $\mu$ | Benefits to the attacker when an attacker intrudes |
| $\beta$ | Penalty to the attacker when an attacker intrudes and a firm invests ($\mu<\beta$) |
| $\alpha$ | Effect of firm's deferral investment on firm and attack's expected benefits ($0\leq\alpha\leq1$) |
| $P_D$ | Probability that CSO obtain $S_H$ through firm's information sharing, given an attacker intrudes |
| $P_F$ | Probability that CSO obtain $S_H$ through firm's information sharing, given an attacker doesn't intrude |
| $S_H$ | The signal of firm's cost saving to be high if a firm invests |
| $S_L$ | The signal of firm's cost saving to be low if a firm invests |
| **Decision variables** | **Explanation** |
| $\rho$ | Probability of firm's investment in the absence of information sharing and deferral option |
| $\psi$ | Probability of attacker's intrusion in the absence of information sharing and deferral option |
| $\rho_{in\_def}$ | Probability of firm's immediate investment in the presence of deferral option |
| $\rho_{no\_def}$ | Probability of firm's non-investment in the presence of deferral option |
| $\rho_{de\_def}$ | Probability of firm's deferral decision in the presence of deferral option |
| $\psi_{def}$ | Probability of attacker's intrusion in the presence of deferral option |
| $\rho_{1in\_sha}$ | Probability of firm's investment, in the presence of information sharing, when the CFO receives $S_H$ |
| $\rho_{2in\_sha}$ | Probability of firm's investment, in the presence of information sharing, when the CFO receives $S_L$ |
| $\psi_{sha}$ | Probability of attacker's intrusion in the presence of information sharing |
| $\rho_{1in\_sha+def}$ | Probability of firm's immediate investment, in the presence of information sharing and deferral option, when the CFO receives $S_H$ |
| $\rho_{2in\_sha+def}$ | Probability of firm's immediate investment, in the presence of information sharing and deferral option, when the CSO obtains $S_L$ |
| $\rho_{1de\_sha+def}$ | Probability of firm's deferral decision, in the presence of information sharing and deferral option, when the CFO receives $S_H$ |
| $\rho_{2de\_sha+def}$ | Probability of firm's deferral decision, in the presence of information sharing and deferral option, when the CFO receives $S_L$ |
| $\psi_{sha+def}$ | Probability of attacker's intrusion, in the presence of information sharing and deferral option |
| **Functions** | **Explanation** |
| $M$ | The firm's expected benefits |
| $A$ | The attacker's expected benefits |

the firm's net benefits are $H$-$C$, and the attacker's net benefits are $\mu-\beta$ when the firm invests and the attacker intrudes. Similarly, a firm's net benefits are $L$-$C$ when it invests and the attacker does not intrude. Neither a firm nor an attacker has no benefit when the firm does not invest and the attacker dose not intrude; a firm could sustain damage when it does not invest and the attacker intrudes. However, the damage has no influence on the conclusion, so we assume that the firm's damage is zero for convenience of calculation. Therefore, whether the attacker intrudes or not, the firm will not benefit if it doesn't invest in it. Regardless of whether the firm invests, the attacker has no benefit if it does not intrude. We assume that the probability of a firm's investment is $\rho$, and the probability of an attacker's intrusion is $\psi$.

More accurate information about investment payoffs can be obtained if a firm defers the investment. However a cybersecurity breach could occur during the deferral process. Therefore, we define $\alpha$ as the effect of a firm's deferral investment on the expected benefits of the

firm and attack. Actually, $\alpha$ is different for the firm and attacker; however, the effect of a firm's deferral investment on the firm and attack's expected benefits is positive correlation. And the difference of $\alpha$ between the firm and attacker has no influence on out conclusion; hence, we assume that the effect of the firm's deferral investment on the expected benefits of the firm and attack is equal for the convenience of calculation. Hence, the firm's net benefits are $\alpha H - C$, and the attacker's net benefits are $\mu - \alpha\beta$ when the firm invests and the attacker intrudes in cybersecurity ($0 < \alpha < 1$).

To facilitate calculation, we assume that only two conditions can be found after deferral. That is, a firm always invests if the attacker intrudes, and the firm does not invest if the attacker does not intrude. Actually, a firm can invest, never invest, or goes on deferring after deferring; however, gradually speaking, the two conditions take the large part of the three choices as the firm gets more useful information about the attacker than before. The conclusion in this paper keep the same if some new parameters are introduced. Additionally, $\alpha H \geq C$, $\mu \geq \alpha\beta$, and $\alpha$ decreased with an increase in deferral time. A firm will not invest if $\alpha H < C$. If an attack intrudes, it must have benefits when the firm defers decision ($\mu \geq \alpha\beta$). This is because the firm gets more useful information with the increase of deferral time, but the attacker also gets more useful information. And the firm has been attacked before it invest if it defers the decision.

$\rho_{in\_def}$, $\rho_{no\_def}$ and $\rho_{de\_def}$ are the firm's probabilities of investing immediately, avoiding up investing (does not invest), and deferring decisions respectively, in the presence of deferral option only. $\psi_{def}$ is the attacker's probability of intrusion in the presence of the deferral option only.

Deferral decisions in cybersecurity aim to decrease uncertainty associated with potential payoff from cybersecurity investments. Information sharing can reduce uncertainty to some extend and the value of deferral option [26]. Accuracy of the information acquired from information-sharing is critical in reducing uncertainty. We will discuss the accuracy by introducing a receiver operating characteristic (ROC) curve [27,28] in a practical example [26], wherein the model is similar to the intrusion detection system (IDS) discussed by Cavusoglu et al. [29] and firewall discussed by Cavusoglu and Raghunathan [30].

## 2.2. Information sharing and ROC curve

In the example [31], 60% of the budget is earmarked for basic cybersecurity activities, and the Chief Security Officer (CSO) is authorized to use these funds. However, the remaining 40% of the cybersecurity budget cannot be spent without approval from the firm's Chief Financial Officer (CFO). The CSO aims to use the remaining portion of the firm's cyber security budget to hire a consulting firm to enhance the cyber security operations of its clients. From the CSO's perspective, hiring a cybersecurity consulting firm now rather than later makes sense, as the CSO bears the ultimate responsibility for actual security breaches. According to advice from the CSO, the CFO can invest immediately (hire a cybersecurity consulting firm), defer their decision, and avoid the investment.

We measure the effectiveness of firms' information sharing through parameters $P_D$ and $P_F$. $P_D$ denotes the probability that SCO obtains an $S_H$ from information sharing when an attacker intrudes, where $S_H$ is the signal of a firm's cost savings to be high ($H$) if it invests. Unlike $S_H$, $S_L$ signals that a firm's cost savings is low ($L$) if it invests. $P_F$ is the probability that an SCO obtains an $S_H$ from information sharing when an attack does not intrude. Similar to an IDS, we consider a CSO that uses a number score $x$ estimated from the information sharing and a threshold value $t$ in the heart of the CSO to determine whether hiring the cybersecurity consulting firm immediately can help the firm gain high-cost savings. For $x > t$, the CSO views the attacker as an attacker who intrudes. It follows that $P_D = \int_t^\infty f_I(x)dx$, $P_F = \int_t^\infty f_N(x)dx$, where

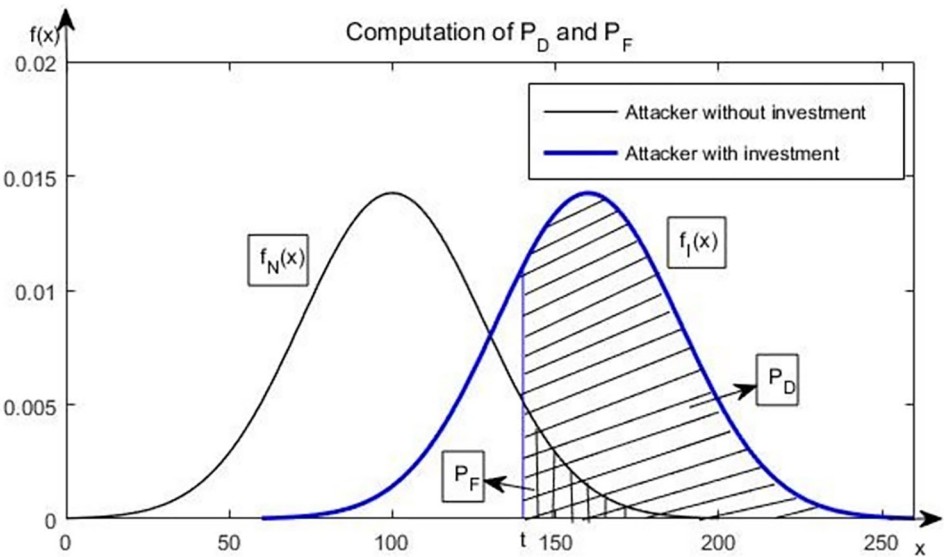

**Fig 1. Probability calculation.**

$f_I(x)$ and $f_N(x)$ are the probability density functions of $x$ for attackers with and without intrusion. Fig 1 illustrates the probability calculation. For a given CSO and information sharing, we capture the relationship between $P_D$ and $P_F$ as $P_F = (P_D)^r$, where $r$ captures the CSO's risk preference. We derived this functional form for the ROC curve as follows:

Number score, which is used to distinguish an attacker with and without intrusion, follows an exponential distribution [32]. If the numerical scores for the attacker who intrudes and does not intrude follow exponential distributions with parameters $\theta_I$ and $\theta_N$, $\theta_N > \theta_I$, respectively. We then write $P_D$ and $P_F$ as follows:

$$P_D = \int_t^\infty \theta_I e^{-(\theta_I x)} dx = e^{-\theta_I t} \tag{1}$$

$$P_F = \int_t^\infty \theta_N e^{-(\theta_N x)} dx = e^{-\theta_N t} \tag{2}$$

where $P_D$ can be expressed as a function of $P_F$, and we write it as $P_F = (P_D)^r$, where $r = \theta_N / \theta_H$ is greater than one. Similar to Cavusoglu et al. [29] in Fig 2, parameter $r$ captures the CSO's risk preference and $P_D > P_F$. The CSO can obtain two types of signals ($S_H$ and $S_L$) from information sharing, and the CFO has two types of investment probability according to the CSO's signal. Table 1 lists all the decision variables and their implications.

## 3. Model analysis

### 3.1. Information sharing

*Lemma 1.* Assuming that the firm's information sharing performs better than the firm's random determination because $P_D > P_F$, the frequency of investment is always higher in the scenario of high-cost saving signal than that of low-cost saving scenario (i.e., $\rho_{1in\_def} \geq \rho_{2in\_def}$, $\rho_{1in\_sha+def} \geq \rho_{2in\_sha+def}$). Additionally, the firm may invest in the $S_L$ *signal scenario only when it completely invests in all $S_H$ signal scenarios.*

We derive a mixed-strategy Nash equilibrium between a firm and an attacker. We used the following probability computations to derive the equilibrium.

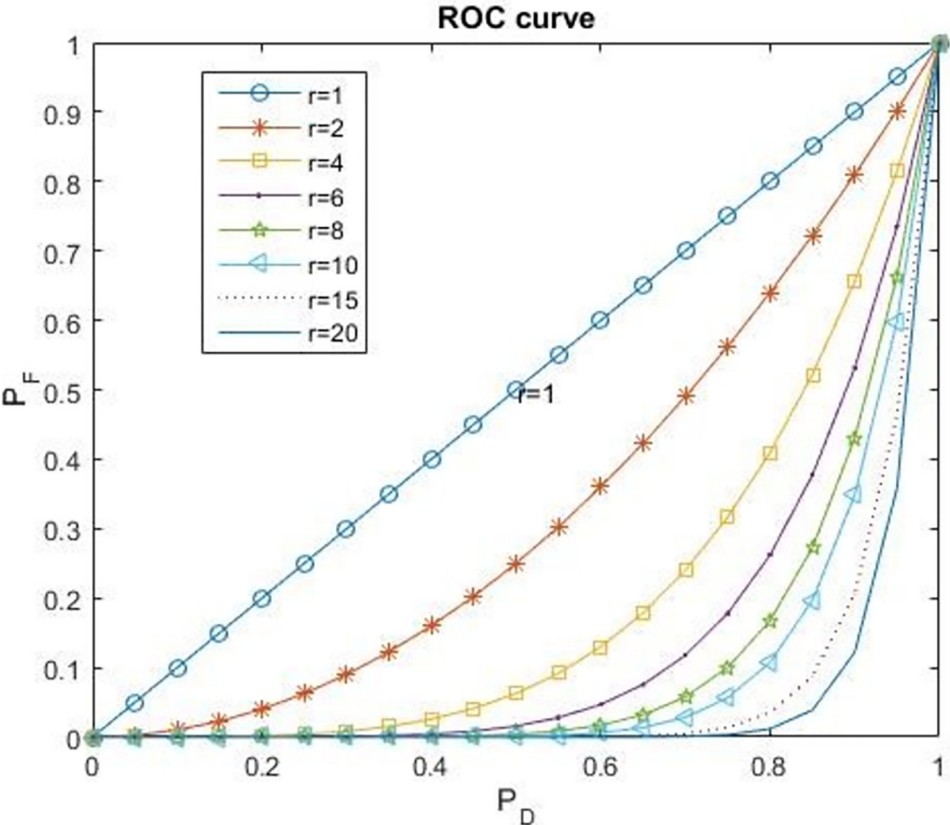

**Fig 2. ROC curve.**

The probabilities of being in the signal of high-cost saving $(S_H)$ and low-cost saving $(S_L)$ scenarios are given by the following:

$$P(S_H) = \psi_{sha}P_D + (1 - \psi_{sha})P_F \tag{3}$$

$$P(S_L) = \psi_{sha}(1 - P_D) + (1 - \psi_{sha})(1 - P_F) \tag{4}$$

According to Bayes' rule, the posterior probabilities of the attacker's intrusion when the firm's CFO receives a signal of high-cost saving and low-cost saving can be calculated as follows:

$$\eta_1 = P(H|S_H) = \frac{P(A\_investment)P(S_H|A\_investment)}{P(S_H)} = \frac{\psi_{sha}P_D}{\psi_{sha}P_D + (1 - \psi_{sha})P_F} \tag{5}$$

$$\eta_2 = P(H|S_L) = \frac{P(A\_investment)P(S_L|A\_investment)}{P(S_L)} = \frac{\psi_{sha}(1 - P_D)}{\psi_{sha}(1 - P_D) + (1 - \psi_{sha})(1 - P_F)} \tag{6}$$

Fig 3 presents the game model. Our model assumes that both the firm's CFO and the attacker have perfect information, all of which are available to both. The firm's expected

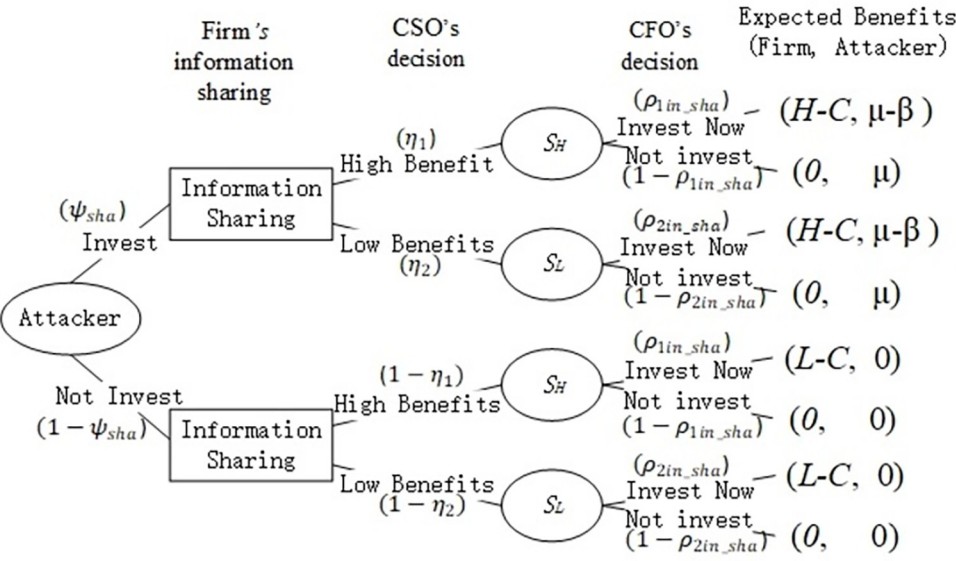

**Fig 3. Game tree with information sharing.**

benefits when receiving a signal of high-cost saving and low-cost saving take the following form:

$$M_H(\rho_{1in_{sha}}, \psi_{sha}) = \eta_1 \rho_{1in_{sha}}(H - C) + (1 - \eta_1)\rho_{1in\_sha}(L - C) \tag{7}$$

$$M_L(\rho_{2in_{sha}}, \psi_{sha}) = \eta_2 \rho_{2in\_sha}(H - C) + (1 - \eta_2)\rho_{2in\_sha}(L - C) \tag{8}$$

Thus, the firm's overall expected benefits is as follows:

$$M_{sha} = M(\rho_{1in_{sha}}, \rho_{2in_{sha}}, \psi_{sha}) = M_H(\rho_{1in\_sha}, \psi_{sha})P(S_H) + M_L(\rho_{2in\_sha}, \psi_{sha})P(S_L) \tag{9}$$

Similarly, the attacker's expected benefits when receiving a signal of high-cost saving and low-cost saving take the following form:

$$A_H(\rho_{1in_{sha}}, \psi_{sha}) = \eta_1[\rho_{1in\_sha}(\mu - \beta) + (1 - \rho_{1in\_sha})\mu] \tag{10}$$

$$A_L(\rho_{2in_{sha}}, \psi_{sha}) = \eta_2[\rho_{2in\_sha}(\mu - \beta) + (1 - \rho_{2in\_sha})\mu] \tag{11}$$

Thus, the attacker's overall expected benefits is as follows:

$$A_{sha} = A(\rho_{1in_{sha}}, \rho_{2in_{sha}}, \psi_{sha}) = A_H(\rho_{1in\_sha}, \psi_{sha})P(S_H) + A_L(\rho_{2in\_sha}, \psi_{sha})P(S_L) \tag{12}$$

We assume that a simultaneous game is played between the firm and attacker. The Nash Equilibrium strategies of this game can be solved wherein neither the firm nor the attacker can improve the game's utility by unilaterally deviating the game's strategy.

**Proposition 1.** When a firm and attacker play a simultaneous game, the optimal frequencies of the firm's investments and the attacker's intrusion for a given condition in the presence of information sharing as follows:

Proof provided in the Appendix.

If $P_D \leq \frac{\mu}{\beta}$, then $\rho_{1in\_sha}^* = 1$, $\rho_{2in\_sha}^* = \frac{\mu - \beta P_D}{\beta(1 - P_D)}$, $\psi_{sha}^* = \frac{(1 - P_F)(C - L)}{(1 - P_D)(H - C) + (1 - P_F)(C - L)}$.

If $\frac{\mu}{\beta} < P_D$, then $\rho_{1in\_sha}^* = \frac{\mu}{\beta P_D}$, $\rho_{2in\_sha}^* = 0$, $\psi_{sha}^* = \frac{P_F(C - L)}{P_D(H - C) - P_F(C - L)}$.

If the firm's information sharing is not considered, the CFO will randomly determines whether an attacker intrudes. Thus, the firm's and attacker's expected benefits take the following form:

$$M = M(\rho, \psi) = \psi\rho(H - C) + (1 - \psi)\rho(L - C) \tag{13}$$

$$A = A(\rho, \psi) = \psi[\rho(\mu - \beta) + (1 - \rho)\mu] \tag{14}$$

According to $\frac{\partial M}{\partial \rho} = 0$ and $\frac{\partial A}{\partial \psi} = 0$, we have $\psi = \frac{C - L}{H - L}$ and $\rho = \frac{\mu}{\beta}$. Hence, the Nash Equilibrium between a firm and an attacker in the absence of information sharing is as follows:

**Proposition 2.** The following mixed strategy profiles constitute the Nash Equilibrium in the given condition in the absence of information sharing.

$$\rho^* = \frac{\mu}{\beta}, \ \psi^* = \frac{C - L}{H - L}$$

To further understand how information sharing affects the strategies of a firm's CFO and attacker, we compare the probability of the firm's investment and the attacker's intrusion in two conditions: in the presence of information sharing and absence of information sharing. We define a firm's investment rate as the probability of an investment in the consulting firm. The firm's investment rate is $\mu/\beta$ in the absence of information sharing (Proposition 2). The firm's investment rate is given by $\rho_{1in\_sha}^* P(S_H) + \rho_{2in\_sha}^* P(S_L)$ in the presence of information sharing, which is equal to the following:

$$\rho_{in\_sha}^* = \frac{\mu}{\beta}\frac{(H - L)P_F}{(H - C)P_D - (C - L)P_F}, \text{ if } \frac{\mu}{\beta} \leq P_D; \text{ and } \rho_{in_{sha}}^* = \frac{(H - L)(\mu/\beta)(1 - P_F) - (H - C)(P_D - P_F)}{(H - L)(1 - P_D) + (C - L)(P_D - P_F)}, \text{ if } \frac{\mu}{\beta} > P_D.$$

**Proposition 3.** (1)When the accuracy ($P_D$) *of information sharing is low, a firm will invest not only in all $S_H$ signal but also in a fraction of $S_L$ signal; when the accuracy ($P_D$) of information sharing is high, the firm won't invest in $S_L$ signal and may only invest in a fraction of $S_H$ signal.*

(2) Information sharing decreases firm's security investment rate.

When the accuracy ($P_D$) of information-sharing is low, a firm invests in all $S_H$ signals because the probability of false positives ($P_F$) is low, and the firm invests in a fraction of $S_L$ signal because the probability of false negatives ($1-P_D$) is high. When the accuracy ($P_D$) of information sharing is high, the firm invests in a fraction of $S_H$ as the probability of false positives ($P_F$) is high, and the firm gives up investing in $S_L$ signal as the probability of false negatives ($1-P_D$) is low.

Information sharing divides the parameter spaces into two regions that help the firm accurately target its investment object. Therefore, to maintain the equilibrium point between the firm and attacker, the firm will decrease its investment rate in the presence of information sharing.

**Table 2. Firm's and attacker's decisions and the relevant benefits when the deferral option is considered.**

| Decisions and probability | | Benefits | |
|---|---|---|---|
| Attacker | Firm | Firm | Attacker |
| Intrude ($\psi_{def}$) | Invest Immediately ($\rho_{in\_def}$) | $H-C$ | $\mu-\beta$ |
| | Defer Decision ($\rho_{de\_def}$) | $\alpha H-C$ | $\mu-\alpha\beta$ |
| | Not Invest ($\rho_{no\_def}$) | 0 | $\mu$ |
| Not intrude ($1-\psi_{def}$) | Invest Immediately ($\rho_{in\_def}$) | $L-C$ | 0 |
| | Defer Decision ($\rho_{de\_def}$) | 0 | 0 |
| | Not Invest ($\rho_{no\_def}$) | 0 | 0 |

## 3.2. Deferral decision and information sharing

We assumed $L \leq C$, $\mu \geq \alpha\beta$, and $\alpha H \geq C$ in the model description in Chapter II. Additionally, we assume that both the firm's CFO and the attacker have perfect information. Table 2 list the decisions of the firm and attacker and the relevant benefits when the deferral option is considered. The firm's and attacker's expected benefits take the following form:

$$M_{def} = M(\rho_{in\_def}, \ \rho_{de\_def}, \ \rho_{no\_def}, \ \psi_{def})$$
$$= \psi_{def}[\rho_{in\_def}(H-C) + \rho_{de\_def}(\alpha H - C)] + (1-\psi_{def})\rho_{in\_def}(L-C) \quad (15)$$

$$A_{def} = A(\rho_{in\_def}, \ \rho_{de\_def}, \ \rho_{no\_def}, \ \psi_{def}) = \psi_{def}\begin{bmatrix} \rho_{in\_def}(\mu-\beta) + \rho_{de\_def}(\mu-\alpha\beta) \\ +\rho_{no\_def}\mu \end{bmatrix} \quad (16)$$

where $\rho_{in\_def} + \rho_{de\_def} + \rho_{no\_def} = 1$. The firm's expected benefits increase in $\rho_{de\_def}$ according to Eq (15), and $\frac{\partial M_{def}}{\partial \rho_{no\_def}} = 0$. Hence, the optimal decision for the firm is $\rho_{no\_def}^* = 0$. Therefore, we have the following:

$$M_{def} = \psi_{def}[\rho_{in\_def}(H-C) + (1-\rho_{in\_def})(\alpha H - C)] + (1-\psi_{def})\rho_{in\_def}(L-C) \quad (17)$$

$$A_{def} = \psi_{def}\begin{bmatrix} \rho_{in\_def}(\mu-\beta) \\ +(1-\rho_{in\_def})(\mu-\alpha\beta) \end{bmatrix} \quad (18)$$

When $\frac{\partial M_{def}}{\partial \rho_{in\_def}} = 0$ and $\frac{\partial A_{def}}{\partial \psi_{def}} = 0$, we have the optimal frequencies of the firm's investment and the attacker's intrusion, for a given condition with the deferral option.

**Proposition 4.** (1)The following mixed strategy profiles constitute the Nash Equilibrium in the given condition of a firm's deferral option.

$$\rho_{in\_def}^* = \frac{\mu-\alpha\beta}{(1-\alpha)\beta}, \ \rho_{de\_def}^* = \frac{\beta-\mu}{(1-\alpha)\beta}, \ \rho_{no\_def}^* = 0, \ \text{and} \ \varphi_{def}^* = \frac{C-L}{(1-\alpha)H + C - L}$$

(2) When a firm can defer its decision, the firm will decrease the probability of immediate investment, and avoid non-investment. That is $\rho_{in\_def}^* < \rho^*$ and $\rho_{no\_def}^* = 0$.

(3) Probability of a firm's deferral decision increases in parameter α, that is $\partial \rho_{de\_def}^* / \partial \alpha > 0$.

For a firm to decide between immediate investment and non-investment is difficult; hence, a deferral decision may be a more sensible decision. Compared with non-investment, deferral decision, which decrease uncertainty, can help firms obtain a part of the benefits. Thus, a firm will decrease the probability of immediate investment and avoid non-investment. Essentially, a

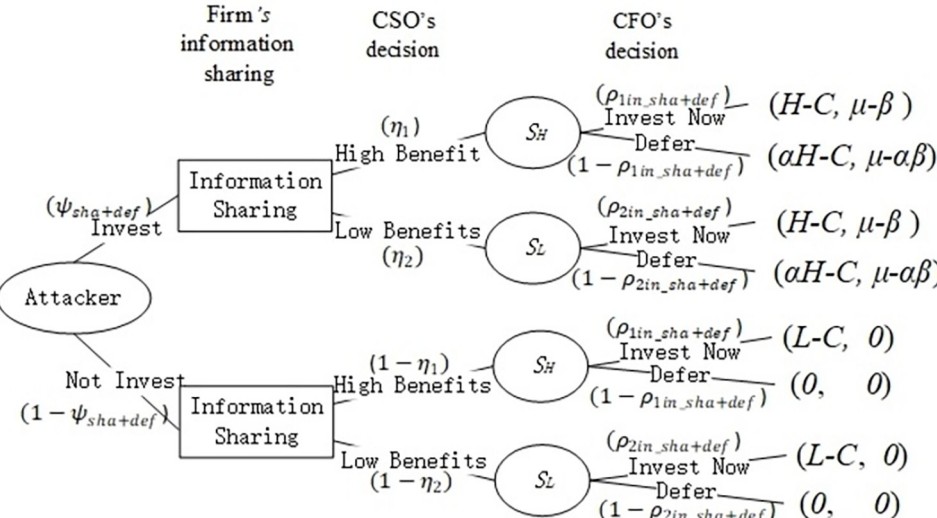

**Fig 4. Game tree in the presence of information sharing and deferral option.**

firm will increase the probability of deferral decision to make a more sensible decision. According to Proposition 4, a firm will avoid non-investment if the firm can defer its decision. Hence, assuming that the firm and attacker are simultaneously playing games, we construct a game tree (Fig 4) when both the firm's information sharing and deferral option are considered.

Similar to the scenario with only the firm's information sharing, if both the firm's information sharing and deferral option are considered, the firm's expected benefits when obtaining a signal of high-cost saving and low-cost saving take the following form:

$$M_H(\rho_{1in\_sha+def}, \psi_{sha+def}) = \eta_1[\rho_{1in\_sha+def}(H - C) + (1 - \rho_{1in\_sha+def})(\alpha H - C)] + (1 - \eta_1)\rho_{1in\_sha+def}(L - C) \quad (19)$$

$$M_L(\rho_{2in\_sha+def}, \psi_{sha+def}) = \eta_2[\rho_{2in\_sha+def}(H - C) + (1 - \rho_{2in\_sha+def})(\alpha H - C)] + (1 - \eta_2)\rho_{2in\_sha+def}(L - C) \quad (20)$$

Thus, the firm's overall expected benefits is

$$M_{sha+def} = M(\rho_{1in_{sha+def}}, \rho_{2in_{sha+def}}, \psi_{sha+def}) = M_H(\rho_{1in\_sha+def}, \psi_{sha+def})P(S_H) + M_L(\rho_{2in\_sha+def}, \psi_{sha+def})P(S_L) \quad (21)$$

Similarly, if both the firm's information sharing and deferral option are considered, the attacker's expected benefits when obtaining a signal of high-cost saving and low-cost saving take the following form:

$$A_H(\rho_{1in\_sha+def}, \psi_{sha+def}) = \eta_1 \left[ \begin{array}{c} \rho_{1in\_sha+def}(\mu - \beta) \\ +(1 - \rho_{1in\_sha+def})(\mu - \alpha\beta) \end{array} \right] \quad (22)$$

$$A_L(\rho_{2in\_sha+def}, \psi_{sha+def}) = \eta_2 \left[ \begin{array}{c} \rho_{2in\_sha+def}(\mu - \beta) \\ +(1 - \rho_{2in\_sha+def})(\mu - \alpha\beta) \end{array} \right] \quad (23)$$

Thus, the attacker's overall expected benefits is as follows:

$$A_{sha+def} = A(\rho_{1in_{sha+def}}, \rho_{2in_{sha+def}}, \psi_{sha+def}) = A_H(\rho_{1in\_sha+def}, \psi_{sha+def})P(S_H) + A_L(\rho_{2in\_sha+def}, \psi_{sha+def})P(S_L) \quad (24)$$

Proof is similar to Proposition 1: For a given scenario with a firm's information sharing and deferral option, we have the optimal frequencies of the firm's investment and the attacker's intrusion.

**Proposition 5.** When a firm and an attacker play a simultaneous game, for a given scenario with both the firm's information sharing and deferral option, the optimal frequencies of the firm's investments and the attacker's intrusion are as follows:

If $P_D \le \frac{\mu-\alpha\beta}{(1-\alpha)\beta}$, then

$$\rho_{1in\_sha+def}^* = 1, \ \rho_{2in\_sha+def}^* = \frac{\mu-\alpha\beta-P_D(1-\alpha)\beta}{(1-P_D)(1-\alpha)\beta}, \ \psi_{sha+def}^* = \frac{(1-P_F)(C-L)}{(1-P_D)(1-\alpha)H+(1-P_F)(C-L)};$$

$$\rho_{1de\_sha+def}^* = 0, \ \rho_{2de\_sha+def}^* = \frac{\beta-\mu}{(1-P_D)(1-\alpha)\beta}.$$

If $\frac{\mu-\alpha\beta}{(1-\alpha)\beta} < P_D$, then

$$\rho_{1in\_sha+def}^* = \frac{\mu-\alpha\beta}{P_D(1-\alpha)\beta}, \ \rho_{2in\_sha+def}^* = 0, \ \psi_{sha+def}^* = \frac{P_F(C-L)}{P_D(1-\alpha)H+P_F(C-L)};$$

$$\rho_{1de\_sha+def}^* = \frac{P_D(1-\alpha)\beta-\mu+\alpha\beta}{P_D(1-\alpha)\beta}, \ \rho_{2de\_sha+def}^* = 1.$$

Similar to the case with only the firm's information sharing, in the scenario with both the firm's information sharing and deferral option, the firm's investment and deferral rates are given by $\rho_{1in\_sha+def}^* P(S_H)+\rho_{2in\_sha+def}^* P(S_L)$ and $\rho_{1de\_sha+def}^* P(S_H)+\rho_{2de\_sha+def}^* P(S_L)$ respectively, which are equal to the following:

$$\rho_{in\_sha+def}^* = 1 - \rho_{de\_sha+def}^*, \ \rho_{de\_sha+def}^* = \frac{\beta-\mu}{(1-\alpha)\beta} \frac{(1-P_D)\psi_{sha+def}^*+(1-P_F)(1-\psi_{sha+def}^*)}{1-P_D} \text{ if } P_D \le \frac{\mu-\alpha\beta}{(1-\alpha)\beta}; \text{ and}$$

$$\rho_{in\_sha+def}^* = \frac{\mu-\alpha\beta}{(1-\alpha)\beta} \frac{P_D\psi_{sha+def}^*+P_F(1-\psi_{sha+def}^*)}{P_D}, \ \rho_{de\_sha+def}^* = 1 - \rho_{in\_sha+def}^* \text{ if } \frac{\mu-\alpha\beta}{(1-\alpha)\beta} < P_D.$$

**Proposition 6.** *(1) When the accuracy ($P_D$) of firm's information sharing is low, the firm will invest immediately not only all $S_L$ signal, but also a fraction of $S_L$ signal. When the accuracy ($P_D$) of firm's information sharing is high, the firm will avoid investing all $S_L$ signal, and in fact, the firm may invest immediately a fraction of $S_H$ signal.*

*(2) After information sharing, the probability of the firm's deferral decision increases for $S_L$ signal, but it decreases for $S_H$ signal. Besides, the firm's overall deferral rate increases after the information sharing.*

*(3) Attacker decreases its probability of intrusion when the accuracy ($P_D$) of the information sharing is high, but it increases the probability when the accuracy ($P_D$) is low.*

Proof is as follow:

When $P_D \le \frac{\mu-\alpha\beta}{(1-\alpha)\beta}$, $\rho_{de\_sha+def}^* > \rho_{de\_def}^*$ because of $\frac{(1-P_D)\psi_{sha+def}^*+(1-P_F)(1-\psi_{sha+def}^*)}{1-P_D} > 1$. Besides, $\rho_{2de\_sha+def}^* > \rho_{de\_def}^* > \rho_{1de\_sha+def}^* = 0$, and $\psi_{sha+def}^* > \varphi_{def}^*$.

When $\frac{\mu-\alpha\beta}{(1-\alpha)\beta} < P_D$, $\rho_{de\_sha+def}^* > \rho_{de\_def}^*$ because of $\frac{P_D\psi_{sha+def}^*+P_F(1-\psi_{sha+def}^*)}{P_D} < 1 \Longrightarrow \rho_{in\_sha+def}^* < \rho_{in\_def}^*$. Besides, $\rho_{1de\_sha+def}^* < \rho_{de\_def}^* < \rho_{2de\_sha+def}^* = 1$, and $\psi_{sha+def}^* < \varphi_{def}^*$.

Explanation of (1) is similar to that of Proposition 3; hence, it is omitted here. A firm's information sharing helps the firm accurately target its investment object. Thus, the firm increases the probability of immediate investment in the $S_H$ signal case (firm decreases the probability of its deferral decision) and decreases the probability of immediate investment in the $S_L$ signal case (firm increases the probability of its deferral decision). The extent of the change in a firm's deferral decision in $S_L$ signal is higher than in the $S_H$ signal, causing overall deferral rate to increase. High deterrence of a firm's information sharing cause the attacker to decrease its intrusion probability when the accuracy ($P_D$) of a firm's information sharing is high. This is opposite when the accuracy is low.

## 4. Value of information sharing and deferral option

### 4.1. Value of information sharing and deferral option with default CSO

After analysing the effect of a firm's information sharing and deferral decision on the firm's and attacker's decisions in Chapter III, we discuss the effect on a firm's expected benefits.

Using Proposition 1 and Eq (9), the firm's expected benefits in the absence of information sharing and deferral decision is $M^* = 0$.

Using Proposition 2 and Eq (13), the firm's expected benefits in the presence of only the firm's information sharing is as follows: $M_{sha}^* = \frac{(P_D - P_F)(H-C)(C-L)}{(1-P_D)(H-C)+(1-P_F)(C-L)}$, if $P_D \leq \frac{\mu}{\beta}$; $M_{sha}^* = 0$, if $P_D > \frac{\mu}{\beta}$.

Using Proposition 4 and Eq (17), the firm's expected benefits in the presence of only the firm's deferral option is $M_{def}^* = \frac{(C-L)(\alpha H-C)}{(1-\alpha)H+(C-L)}$.

Using Proposition 5 and Eq (21), the firm's expected benefits in the presence of the firm's information sharing and deferral decision is $M_{sha+def}^* = \frac{(C-L)[(1-P_D)\alpha H+(P_D-P_F)H-(1-P_F)C]}{(1-P_D)(1-\alpha)H+(1-P_F)(C-L)}$, if $P_D \leq \frac{\mu-\alpha\beta}{(1-\alpha)\beta}$; $M_{sha+def}^* = \frac{P_F(C-L)(\alpha H-C)}{P_D(1-\alpha)H+P_F(C-L)}$, if $P_D > \frac{\mu-\alpha\beta}{(1-\alpha)\beta}$.

***Proposition 7.*** (1) Firm's information sharing can increase a firm's expected benefits only when the accuracy is low, and the firm's expected benefits increases in the accuracy ($P_D$).

(2) Deferral decision can increase a firm's expected benefits, and the firm's expected benefits increase in parameter α.

(3) Firm's information sharing can improve the effect of deferral decision when the accuracy ($P_D$) is low but weaken the effect when the accuracy ($P_D$) is high.

The proof is as follows:

When $P_D \leq \frac{\mu}{\beta}$, $\frac{\partial M_{sha}^*}{\partial P_D} > 0$. $M_{def}^* > M^*$, $\frac{\partial M_{def}^*}{\partial \alpha} > 0$.

When $P_D \leq \frac{\mu-\alpha\beta}{(1-\alpha)\beta}$, $M_{sha+def}^* - M_{def}^* = \frac{(C-L)(1-\alpha)H(P_D-P_F)(H-L)}{[(1-P_D)(1-\alpha)H+(1-P_F)(C-L)][(1-\alpha)H+C-L]} > 0$;

When $P_D > \frac{\mu-\alpha\beta}{(1-\alpha)\beta}$, $M_{sha+def}^* - M_{def}^* = \frac{-(C-L)(\alpha H-C)(1-\alpha)H(P_D-P_F)}{[P_D(1-\alpha)H+P_F(C-L)][(1-\alpha)H+C-L]} < 0$.

Low accuracy ($P_D$) has a low deterrence to the attacker, causing the attacker's intrusion probability to become high and stable. Therefore, a firm's decision in Propositions 1 and 5 can make the firm increase its benefits. A high accuracy ($P_D$) has a high deterrence to the attacker, causing the attacker's intrusion probability to become low and unstable. Therefore, the firm's decision in Propositions 1 and 5 cannot increase its benefits.

Deferring cybersecurity investment can cause some cybersecurity breaches. However, it decreases uncertainty, and the firm can make a more sensible decision; hence the firm's expected benefits increase. With an increase in parameter $\alpha$, the negative effect of deferral investment on the firm's benefits decreases; hence, the firm's expected benefits increase in parameter α. Both the firm's deferral decision and information sharing aim to decrease the uncertainty associated with the firm's potential costs.

When the accuracy ($P_D$) of a firm's information sharing is low, the firm will increase the probability of the deferring decision in Proposition 5. Hence, the firm will decrease the probability of investing in the $S_L$ single. Investment in $S_L$ singles is inefficient in increasing the firm's expected benefits; hence, when the accuracy is low, the firm's information sharing improves the effect of the firm's deferring decision. When the accuracy ($P_D$) of a firm's information sharing is high, the firm will decrease the probability of the deferring decision in Proposition 5. Hence, the firm will increase the probability of investing in the $S_L$ single. Therefore, when the accuracy is high, the firm's information sharing weakens the effect of the firm's decision to defer.

## 4.2. Value of information sharing and deferral option with optional CSO

Proposition 7 indicates that by choosing $P_D$ (or choosing CSO), the CFO can determine how to operate. In the presence of a firm's information sharing and deferral decision, a comparison of the firm's benefits in the equilibrium regions demonstrates that the firm realizes high benefits when $P_D \leq (\mu - \alpha\beta)/[(1-\alpha)\beta]$. Consequently, the CFO will choose the value of $P_D$ to realize $P_D \leq (\mu - \alpha\beta)/[(1-\alpha)\beta]$. Next, the firm should decide where to lie in this region. When $P_D \leq (\mu - \alpha\beta)/[(1-\alpha)\beta]$ writing the benefit as a function of $P_D$ and taking the first derivative yields the following:

$$\frac{\partial M^*_{sha+def}}{\partial P_D} > 0. \tag{25}$$

Proof: $\frac{\partial (M^*_{sha+def} - M^*_{def})}{\partial P_D} > 0$ and $\frac{\partial M^*_{def}}{\partial P_D} = 0$.

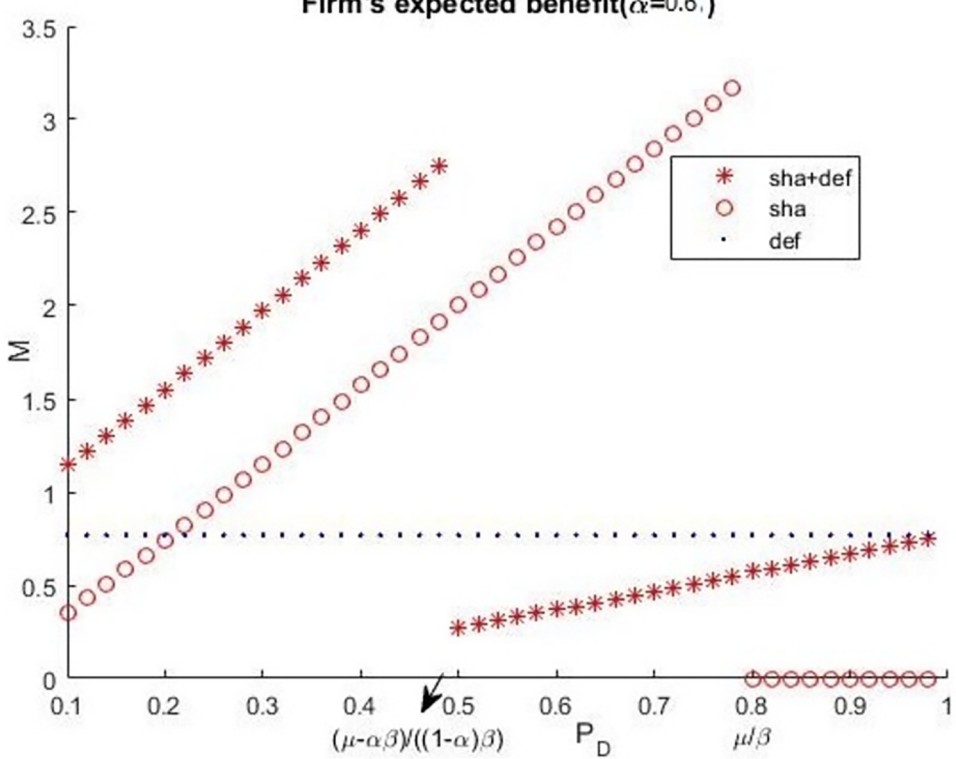

**Fig 5. Firm's expected benefits ($\alpha = 0.6$).**

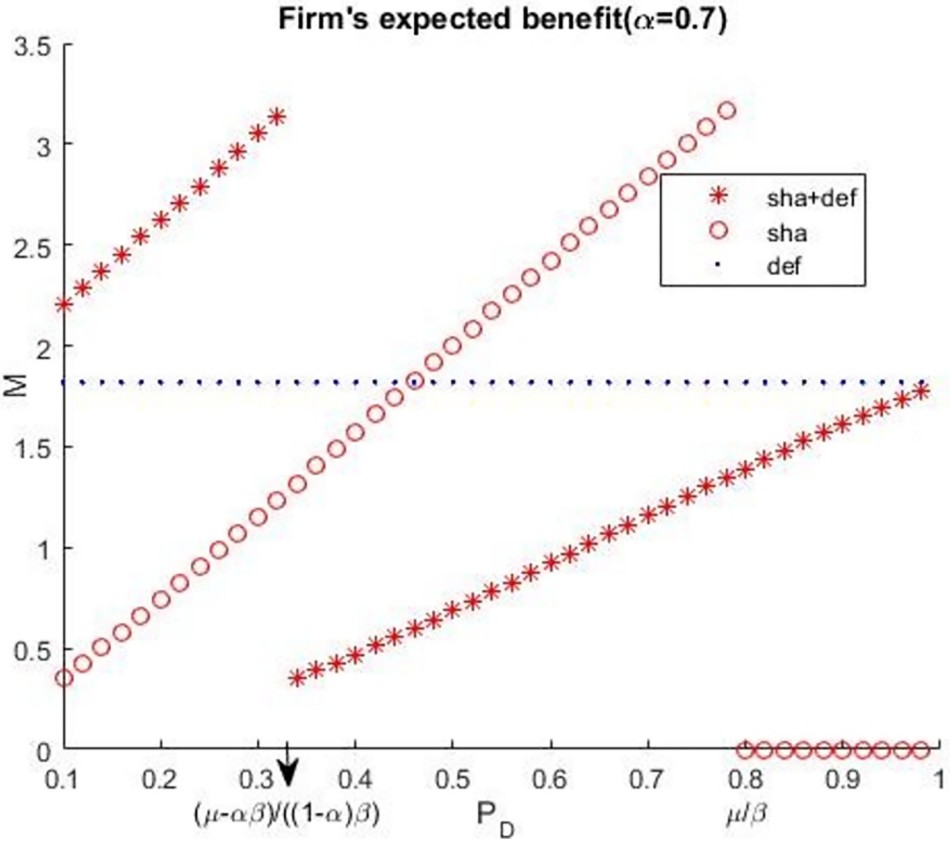

**Fig 6. Firm's expected benefits ($\alpha = 0.7$).**

This derivative implies that the CFO should maximize choose the maximum $P_D$ when $P_D \leq (\mu - \alpha\beta)/[(1-\alpha)\beta]$. Thus, the optimal $P_D$ is $P_D^* = (\mu - \alpha\beta)/[(1-\alpha)\beta]$ in this region, with a firm's deferral option and information sharing. Similarly, a firm in the absence of its deferral option should maximize choose maximum $P_D$ when $P_D \leq \mu/\beta$. Hence, if only a firm's information sharing is considered, $P_D^* = \mu/\beta$. A firm's information sharing has a negative effect on its deferral decision when accuracy ($P_D$) is high; hence, it is omitted here.

To further analyse the effect of a firm's information sharing and deferral option, we calculated the firm's expected benefits for information sharing, deferral decision, and both information sharing and deferral decision. These are a number of examples. Let $C = 10$, $H = 20$, $L = 5$, r = 3, $\mu = 0.8$, $\beta = 10$. Fig 5 presents the condition of $\alpha = 0.6$, Fig 6 presents the condition of $\alpha = 0.7$.

Figs 5 and 6 illustrate Proposition 7. Additionally, when accuracy ($P_D$) is low, the firm's optimal decision combine its information sharing and deferral decision. When accuracy ($P_D$) is between $(\mu - \alpha\beta)/[(1-\alpha)\beta]$ and $\mu/\beta$, the firm's optimal decision is only its information sharing. When accuracy ($P_D$) is high enough, which is higher than $\mu/\beta$, the firm's optimal decision is only its deferral decision. Accuracy ($P_D$) is related to the risk preference of the CSO, and the probability of an attacker's breach is related to the time of deferral and accuracy ($P_D$). Thus, a firm should analyses the risk preference of CSO, the accuracy of information sharing, and the time of deferral before making its decisions in cyber security investment.

## 5. Conclusion

Based on a game framework between a firm and an attacker, we determined the effect of the firm's information sharing and deferral option on a firm's expected benefits. We divided the parameter space of the Chief Financial Officer's decision into different regions by different accuracy of information sharing. Notably, after a firm's information sharing, the probability of a firm's deferral decision decreases for a high-benefit signal but increases for a low-benefit signal. Additionally, a firm's overall deferral rate increases after information sharing. Moreover, a firm's information sharing can improve the effect of its deferral decision on its expected benefits when the accuracy of information sharing is low, but weakens the effect when the accuracy of information sharing is high.

This article includes a deferral option; however, deferral time is not considered. In fact, the effect of deferral option on a firm's expected benefits is closely related to deferral time. Deferral time will be an interesting issue for future research. To facilitate calculation, we assume that neither a free-ride nor a cost for a firm's information sharing. This will be considered in the future, and the relationship between the probability of accuracy and false positives will be an interesting but challenging issue.

## Supporting information

**S1 Appendix.**
(DOCX)

## Author Contributions

**Conceptualization:** Chuanxi Cai.

**Data curation:** Chuanxi Cai.

**Formal analysis:** Chuanxi Cai.

**Funding acquisition:** Chuanxi Cai, Liurong Zhao.

**Investigation:** Chuanxi Cai.

**Methodology:** Chuanxi Cai.

**Project administration:** Chuanxi Cai.

**Resources:** Chuanxi Cai.

**Software:** Chuanxi Cai.

**Supervision:** Chuanxi Cai.

**Validation:** Chuanxi Cai, Liurong Zhao.

**Visualization:** Chuanxi Cai.

**Writing – original draft:** Chuanxi Cai.

**Writing – review & editing:** Liurong Zhao.

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
