## [Decision Letter · Decision Letter 0]

6 Nov 2022

PONE-D-22-24920Information sharing and deferral option in cyber security investmentPLOS ONE

Dear Dr. Cai,

Thank you for submitting your manuscript to PLOS ONE. After careful consideration, we feel that it has merit but does not fully meet PLOS ONE’s publication criteria as it currently stands. Therefore, we invite you to submit a revised version of the manuscript that addresses the points raised during the review process.

Dear authors,

I have now heard back from an expert reviewer on your paper entitled Information Sharing and Deferral Option in Cyber Security Investment. The reviewer considers you work is an interesting contribution. The reviewer has a couple of comments about the “deferral option” that you should address. I agree with him/her, and you must take in account carefully to all points raised in the report.

Moreover, the reviewer mentioned that one criteria is not in line with the publication criteria of PLOS One. I do agree, and I expect a better improvement that suggested in the reviewer’s report. You must improve the written English in the manuscript to reach the required standard: "an intelligible fashion and written in standard English". This is a major point to address.

Irrespective of the direction you eventually choose I registered the submission as major revision. PLOS One often has short deadlines. You should let the journal managers know that you need more time (if you do; the deadlines are not useful for theoretical economic work). Extension of the auto-deadline is fine with me.

Sincerely,

Olivier Bos

We look forward to receiving your revised manuscript.

Kind regards,

Olivier Bos

Academic Editor

PLOS ONE

Journal Requirements:

"This work was supported in part by the Science and Technology Innovation Fund (163060171) and in part by the General Program in philosophy and Social Sciences (2022SJYB0124).

The Science and Technology Innovation Fund (163060171) and the General Program in philosophy and Social Sciences (2022SJYB0124) are all received by Chuanxi Cai."

Reviewers' comments:

Reviewer's Responses to Questions

**Comments to the Author**

1. Is the manuscript technically sound, and do the data support the conclusions?

Reviewer #1: Yes

2. Has the statistical analysis been performed appropriately and rigorously? 

Reviewer #1: Yes

3. Have the authors made all data underlying the findings in their manuscript fully available?

Reviewer #1: Yes

4. Is the manuscript presented in an intelligible fashion and written in standard English?

Reviewer #1: No

5. Review Comments to the Author

Reviewer #1: September 16, 2022

Report PONE-D-22-24920

1. The article provides an interesting analysis of information sharing and the deferral option in security investment.

2. One main contribution is to include the deferral option, which is not much analyzed in the literature.

3. A native reader is needed. The article ignores most rules on singular/plural, when to use “the,” etc.

4. The article should specify that the analysis is about information sharing between firms, not information sharing between attackers.

5. The article considers only one firm, abstracting away that information sharing actually occurs between at least two firms (or between at least two attackers). This limitation should be discussed. Especially, how can the results be realistic when only one firm is analyzed?

6. The first sentence in the abstract says “information security investment strategies.” The article should specify more clearly exactly which number of strategies are available for each player.

7. The deferral option should be defined more clearly. What is being deferred? The reader quickly realizes that the firm’s security investment (singular or plural?) may or may not be deferred. Can information sharing be deferred? Why does the reader have to search forever to find out whether or not information sharing can be deferred?

8. The authors should ensure that all the articles in the reference list, checking one by one, are cited inside the article.

9. The alpha is crucial, and should probably be different for the firm and the attacker.

10. The article’s focus on the deferral option pertains to whether the firm is proactive by investing early, or retroactive by investing later after a deferral. Comparison of the approach and results with the following article seems useful: Hausken, K. (2018), “Proactivity and Retroactivity of Firms and Information Sharing of Hackers,” International Game Theory Review 20, 1, 1750027, doi: 10.1142/S021919891750027X.

11. More generally, comparing the approach and results with the articles in the reference list should be made more thoroughly, accounting for the fact that the article considers only one firm.

12. The weak abstract should be strengthened substantially, listing and discussing the results, emphasizing the contribution relative to the literature, etc. Conclusions can be written without parameters.

6. PLOS authors have the option to publish the peer review history of their article (what does this mean?). If published, this will include your full peer review and any attached files.

Reviewer #1: **Yes: **Kjell Hausken

---

## [Author Response · Author response to Decision Letter 0]

25 Nov 2022

Dear Reviewers:

Thank you for your review, we have amended this article according to your advice. All the question and answer are as follows:

1. The article provides an interesting analysis of information sharing and the deferral option in security investment.

2. One main contribution is to include the deferral option, which is not much analyzed in the literature.

Answer: we have added the analysis of the deferral option in the third paragraph of introduction.

3. A native reader is needed. The article ignores most rules on singular/plural, when to use “the,” etc.

Answer: we have checked and fixed all the rules on singular/plural, when to use “the,” etc.

4. The article should specify that the analysis is about information sharing between firms, not information sharing between attackers.

5. The article considers only one firm, abstracting away that information sharing actually occurs between at least two firms (or between at least two attackers). This limitation should be discussed. Especially, how can the results be realistic when only one firm is analyzed?

Answer to question 4 and 5: This study assumes that firms have joined an industry-specific information-sharing group. No charges are incurred for joining this information-sharing group, providing that a firm is willing to share cybersecurity-related information with the group’s members (i.e., free-riders are excluded from this group). Based on the agreement, all firms report detailed information to the group’s members on their actual cybersecurity breaches and the steps taken to prevent and respond to cybersecurity breaches. Hence, the study constructs a game model between a firm and an attacker; however, the firm can enjoy information sharing. In this study, information sharing occurs between the members of the industry-specific information-sharing group. Additionally, the analysis in this study concerns information sharing between firms, not information sharing between attackers. All of these has been specified in the previous section of the model description.

6. The first sentence in the abstract says “information security investment strategies.” The article should specify more clearly exactly which number of strategies are available for each player.

Answer: A firm has three strategies: immediate investment, no investment, and decision-making after deferral. An attacker has only two strategies: intrusion and no intrusion. All of these has been specified in the previous section of the model description.

7. The deferral option should be defined more clearly. What is being deferred? The reader quickly realizes that the firm’s security investment (singular or plural?) may or may not be deferred. Can information sharing be deferred? Why does the reader have to search forever to find out whether or not information sharing can be deferred?

Answer: The deferral option is a strategy of firm’s security investment, not the strategy of firm’s information sharing. That is, firm’s cybersecurity investment can be deferred, firm’s information sharing cannot be deferred. All of these has been specified in the previous section of the model description.

8. The authors should ensure that all the articles in the reference list, checking one by one, are cited inside the article.

Answer: All the articles in the reference list has been checked. We have deleted one of the articles that is not cited inside the article.

9. The alpha is crucial, and should probably be different for the firm and the attacker.

Answer: α is different for the firm and the attacker, however, the effect of firm’s deferral investment on firm and attack’s expected benefits is positive correlation. Additionally, the difference of α between firm and attacker has no influence on our conclusion; hence, we assume the effect of firm’s deferral investment on the expected benefits of firm and attack is equal for the convenience of calculation. We have illustrated it in the Model Description.

10. The article’s focus on the deferral option pertains to whether the firm is proactive by investing early, or retroactive by investing later after a deferral. Comparison of the approach and results with the following article seems useful: Hausken, K. (2018), “Proactivity and Retroactivity of Firms and Information Sharing of Hackers,” International Game Theory Review 20, 1, 1750027, doi: 10.1142/S021919891750027X.

Answer: Similar to the proactive and retroactive defences in Hausken (2018), firm’s defence is proactive if it invests immediately, or retroactive if it invests later after a deferral.

The information sharing in this article is the information sharing between firms, but the information sharing in Hausken (2018) is the information sharing between hackers.

This paper investigates the effect of information sharing and deferral option on a firm’s expected benefits, and the firm and the attacker are simultaneously playing game. Notably, information sharing can improve the effect of deferral decision on a firm’s expected benefits when the accuracy of information sharing is low but weaken the effect when the accuracy of information sharing is high. Hausken (2018) analysis the interplay between the information sharing of hackers and the defense strategies of firms, and the game is four-period games. Notably, firm prefers to deter the first disadvantaged hacker when the two hackers benefit substantially from information sharing, reputation gain, or the second player is advantaged. All of these has been specified in the third and fourth paragraph of introduction.

11. More generally, comparing the approach and results with the articles in the reference list should be made more thoroughly, accounting for the fact that the article considers only one firm.

Answer: this article considers only one firm; however, this article assume the firm has joined an industry-specific information-sharing group, and there is no charges are incurred for joining this information-sharing group, providing that a firm is willing to share cybersecurity-related information with the group’s members (i.e., free-riders are excluded from this group). The relationship between firms is not considered, but the firm can share information with other firms in the group. Therefore, the results in this paper can be applied to the condition with two or more firms.

12. The weak abstract should be strengthened substantially, listing and discussing the results, emphasizing the contribution relative to the literature, etc. Conclusions can be written without parameters.

Answer: we have added the contribution of this paper relative to the literature ate the end of the abstract. The parameters in the conclusions are replaced with the definition.

Finally, we have employed a professional scientific editing service (Editage) for this study’s language usage, spelling, and grammar. We deeply appreciate your consideration of our manuscript, and give us good reviews. If you have any queries, please do not hesitate to contact us.

Thank you and best regards. 

Yours sincerely

Chuanxi Cai

---

## [Decision Letter · Decision Letter 1]

13 Dec 2022

PONE-D-22-24920R1Information sharing and deferral option in cybersecurity investmentPLOS ONE

Dear Dr. Cai,

Thank you for submitting your manuscript to PLOS ONE. After careful consideration, we feel that it has merit but does not fully meet PLOS ONE’s publication criteria as it currently stands. Therefore, we invite you to submit a revised version of the manuscript that addresses the points raised during the review process.

 I have now carefully checked your review responses and read the revised version in detail again. The reviewer and I are happy about your revision. Yet you ignored the reviewer request about the command of knowledge in English. I urge you to proceed and follow his/her advise. I cannot accept the paper without this improvement. 

We look forward to receiving your revised manuscript.

Kind regards,

Olivier Bos

Academic Editor

PLOS ONE

Journal Requirements:

Reviewers' comments:

Reviewer's Responses to Questions

**Comments to the Author**

1. If the authors have adequately addressed your comments raised in a previous round of review and you feel that this manuscript is now acceptable for publication, you may indicate that here to bypass the “Comments to the Author” section, enter your conflict of interest statement in the “Confidential to Editor” section, and submit your "Accept" recommendation.

Reviewer #1: (No Response)

2. Is the manuscript technically sound, and do the data support the conclusions?

Reviewer #1: Yes

3. Has the statistical analysis been performed appropriately and rigorously? 

Reviewer #1: Yes

4. Have the authors made all data underlying the findings in their manuscript fully available?

Reviewer #1: Yes

5. Is the manuscript presented in an intelligible fashion and written in standard English?

Reviewer #1: No

6. Review Comments to the Author

Reviewer #1: The authors have addressed my concerns, but ignored my request for a native reader. For example, the authors do not know when to use “the” and “a”, mixes past tense and present tense in the literature review, several typos exist, and the article ignores rules on singular/plural. For example, after equation (6) the following sentence is presented: “Fig. 3 indicate the game model.” Since Fig. 3 is singular, it should be “Fig. 3 indicates the game model.” or, which is better, “Fig. 3 presents the game model.”

7. PLOS authors have the option to publish the peer review history of their article (what does this mean?). If published, this will include your full peer review and any attached files.

Reviewer #1: **Yes: **Kjell Hausken

---

## [Author Response · Author response to Decision Letter 1]

18 Dec 2022

Dear Reviewers:

Thank you for your review, we have amended this article according to your advice. All the question and answer are as follows:

1. The authors have addressed my concerns, but ignored my request for a native reader. For example, the authors do not know when to use “the” and “a”, mixes past tense and present tense in the literature review, several typos exist, and the article ignores rules on singular/plural. For example, after equation (6) the following sentence is presented: “Fig. 3 indicate the game model.” Since Fig. 3 is singular, it should be “Fig. 3 indicates the game model.” or, which is better, “Fig. 3 presents the game model.”

Answer: we have checked and fixed all the rules on “the” and “a”, past tense and present tense, singular/plural. Additionally, several typos are also fixed. We deeply appreciate your consideration of our manuscript, and give us good reviews. If you have any queries, please do not hesitate to contact us.

Thank you and best regards. 

Yours sincerely

Chuanxi Cai

---

## [Editor Report · Decision Letter 2]

20 Jan 2023

Information sharing and deferral option in cybersecurity investment

PONE-D-22-24920R2

Dear Dr. Cai,

We’re pleased to inform you that your manuscript has been judged scientifically suitable for publication and will be formally accepted for publication once it meets all outstanding technical requirements.

Kind regards,

Olivier Bos

Academic Editor

PLOS ONE
---

## [Editor Report · Acceptance letter]

26 Jan 2023

PONE-D-22-24920R2 

Information sharing and deferral option in cybersecurity investment 

Dear Dr. Cai:

I'm pleased to inform you that your manuscript has been deemed suitable for publication in PLOS ONE. Congratulations! Your manuscript is now with our production department. 

Kind regards, 

on behalf of

Dr. Olivier Bos 

Academic Editor

PLOS ONE